# Engineered transfer RNAs for suppression of premature termination codons

John D. Lueck[1], Jae Seok Yoon[2], Alfredo Perales-Puchalt [3], Adam L. Mackey[4], Daniel T. Infield[4], Mark A. Behlke [5], Marshall R. Pope[4], David B. Weiner[3], William R. Skach[2,6], Paul B. McCray Jr.[7] & Christopher A. Ahern [4]

Premature termination codons (PTCs) are responsible for 10–15% of all inherited disease. PTC suppression during translation offers a promising approach to treat a variety of genetic disorders, yet small molecules that promote PTC read-through have yielded mixed performance in clinical trials. Here we present a high-throughput, cell-based assay to identify anticodon engineered transfer RNAs (ACE-tRNA) which can effectively suppress in-frame PTCs and faithfully encode their cognate amino acid. In total, we identify ACE-tRNA with a high degree of suppression activity targeting the most common human disease-causing nonsense codons. Genome-wide transcriptome ribosome profiling of cells expressing ACE-tRNA at levels which repair PTC indicate that there are limited interactions with translation termination codons. These ACE-tRNAs display high suppression potency in mammalian cells, *Xenopus* oocytes and mice in vivo, producing PTC repair in multiple genes, including disease causing mutations within cystic fibrosis transmembrane conductance regulator (*CFTR*).

---

[1] Department of Physiology and Pharmacology, University of Rochester School of Medicine and Dentistry, Rochester, NY 14642, USA. [2] CFFT Lab, Cystic Fibrosis Foundation Therapeutics, Lexington 02421 MA, USA. [3] The Wistar Institute, Philadelphia 19104 PA, USA. [4] Department of Molecular Physiology and Biophysics, Iowa Neuroscience Institute, Carver College of Medicine, University of Iowa, Iowa City, IA 52242, USA. [5] Integrated DNA Technologies Inc., Coralville, IA 52241, USA. [6] Cystic Fibrosis Foundation, Bethesda 20814 MD, USA. [7] Stead Family Department of Pediatrics, Pappajohn Biomedical Institute, University of Iowa, Iowa City, IA 52242, USA. Correspondence and requests for materials should be addressed to J.D.L. (email: john_lueck@urmc.rochester.edu) or to C.A.A. (email: christopher-ahern@uiowa.edu)

Premature termination codons (PTCs) arise from single nucleotide mutations that convert a canonical triplet nucleotide codon into one of three stop codons, e.g., TAG, TGA, or TAA. PTCs are often more deleterious than missense mutations because they result in the loss of protein expression. Additionally, mRNA abundance is reduced through nonsense-mediated decay (NMD) and in some cases, truncated proteins may have a dominant negative function[1–3]. Therefore, it is not surprising that PTCs are associated with severe disease phenotypes, including cystic fibrosis[4], Duchenne muscular dystrophy, spinal muscular atrophy[5], infantile neuronal ceroid lipofuscinosis[6], β-thalassemia[7], cystinosis[8], X-linked nephrogenic diabetes insipidus[9], Hurler syndrome[10], Usher syndrome[11], and polycystic kidney disease. Nonsense mutations are also found within the tumor suppressor genes *p53* and *ATM*[12], further implicating their role in disease. Amino acid codons most vulnerable to PTC conversion are those with a single nucleotide substitution from a stop codon: tryptophan, tyrosine, cysteine, glutamic acid, lysine, glutamine, serine, leucine, arginine, and glycine (Supplementary Figure 1). As such, PTCs represent a unique constellation of diseases which afflict over 30 million people worldwide, accounting for 10–15% of all genetic diseases[13].

Small molecules, such as aminoglycosides[14], dipeptides[15], and oxadiazoles[16] promote the "read-through" or "suppression" of nonsense mutations. These compounds are effective in model organisms[17,18], mammalian cell lines[19], and some animal disease models[16,20]. However, this approach results in the encoding of a near-cognate amino acid[21], effectively generating a missense mutation at the PTC, which itself may have deleterious effects on protein folding, trafficking, and function. Furthermore, aminoglycosides are oto- and nephrotoxic[22], and the first-in-class oxadiazole, Ataluren, displayed unexpectedly low efficacy in patient populations (ACT DMD Phase 3 clinical trial, NCT01826487; ACT CF, NCT02139306). Recent and ongoing advances in CRISPR/Cas9-mediated genome editing provides potentially a permanent solution for diseases resulting from nonsense mutations. However, aspects of this technology impart hurdles for its rapid use as a therapeutic[23,24] including cell type specific delivery, the efficiency of homologous recombination, and the frequency of off-target editing, and these challenges are not limited to the requirement of "precision" or "personalized" diagnostics for each mutation based on the context of each patient's genetic variability.

We sought to identify a PTC repair approach that displays the versatility of a small molecule therapeutic and the precision of gene editing. We investigated tRNAs to fulfill these criteria, whereby their anticodons have been engineered via mutagenesis to recognize and suppress UGA, UAA, or UAG PTC codons. In order to be effective, the anticodon edited tRNAs, aka ACE-tRNAs, should still be recognized by the endogenous translation cellular machinery, including the aminoacyl-tRNA synthetase for charging the ACE-tRNA with their cognate amino acid and the eukaryotic elongation factor 1α (eEF-1α) for delivery of the charged tRNA to the ribosome, Fig. 1a. Such suppressor tRNAs have been suggested previously[25], and shown in a limited manner, to rescue in frame stop codons associated with β-thalassemia[26], xeroderma pigmentosum[27], and PTC reporter genes[28,29].

Here we show that an anti-codon editing approach is generalizable to multiple tRNA gene families, indicating that many annotated tRNA are biologically viable. Further, we demonstrate that anti-codon edited suppressor tRNA encode their cognate amino acid, lack significant interactions with termination stop codons and are efficacious in vivo to suppress PTC. In total, the data support the possibility that such engineered tRNA satisfy the broad requirement for coverage of disease-causing PTCs and thus represent a promising new class of RNA therapeutic agent.

## Results

**A streamlined HTS system for ACE-tRNA identification.** The rationale of this study is rooted in the observation that there are multiple tRNA genes with unique sequences (isodecoders) for a given cognate amino acid (isoacceptors), leading to >400

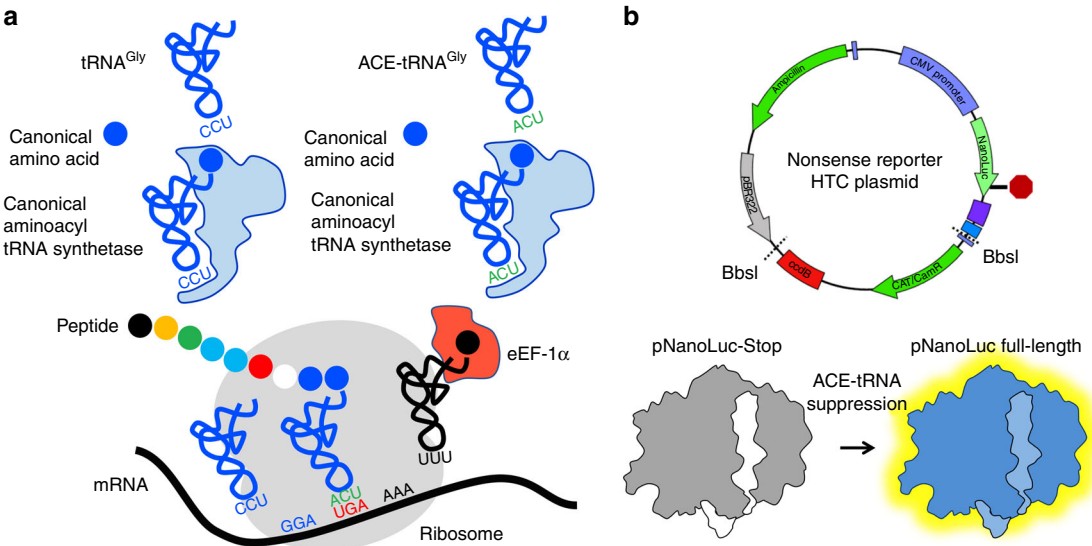

**Fig. 1** A nonsense mutation suppression screen to identify candidate anticodon edited tRNAs (ACE-tRNAs). **a** Schematic illustrates requisite interactions of ACE-tRNAs with translational machinery. Following delivery, ACE-tRNAs are recognized by an endogenous aminoacyl-tRNA synthetase (blue shape) and charged (aminoacylated) with their cognate amino acid (blue circle). The aminoacylated ACE-tRNA is recognized by the endogenous elongation factor 1-alpha (red shape), which protects the ACE-tRNA from being de-acylated and delivers the aminoacyl ACE-tRNA to the ribosome (light gray shape) for suppression of a premature termination codon, in this instance UGA. **b** Individual ACE-tRNAs were cloned into the high throughput cloning nonsense Reporter plasmid using Golden Gate paired with CcdB negative selection. The all-in-one plasmid contains the NLuc luciferase reporter with either a UGA, UAG, or UAA PTC at p.162 between the enzymatic large bit and requisite C-terminal small bit

tRNAs annotated in the human genome (http:lowelab.ucsc.edu/GtRNAdb/)[30,31]. We first examined tRNA genes to identify individual ACE-tRNAs which retain suppression efficacy of PTCs in mammalian cells. In order to maximize sequence coverage, we generated an all-in-one cDNA plasmid that supports both high-throughput cloning (HTC) of ACE-tRNAs and quantitative measurement of PTC suppression using luminescence following delivery to mammalian cells (Fig. 1b). ACE-tRNA sequences were cloned as dsDNA oligos (duplexed oligos) into the HTC plasmid using Golden Gate cloning[32] paired with ccdB negative selection[33]. While the Golden Gate ligation process in extremely efficient, a very low abundance of parent plasmids will inevitably remain in the plasmid population, necessitating colony picking strategies and imparting a bottleneck to the HTC process. We included a ccdB cassette, a potent inhibitor of bacteria growth and proliferation, in the parent plasmid Golden Gate cloning site (Fig. 1b), so that following transformation of a plasmid population that includes parent plasmid only plasmids with successful ACE-tRNA gene ligation will survive. This strategy streamlined the HTC process and produced ~100% cloning efficiency. ACE-tRNA suppression efficiency was read out from a split NanoLuc luciferase (NLuc) NanoBiT platform whereby the PTC of interest (UGA, UAA, or UAG) was introduced in-frame at the junction between the large bit and small bit domains, Fig. 1b[34], using a 96-well format and normalized to background obtained in NLuc-PTC expressing cells. Twenty-one glycine ACE-tRNAs were first evaluated for suppression of the UGA PTC (Fig. 2, top left, column 1) (violet). A majority of the ACE-tRNA$^{Gly}$ sequences failed to suppress the UGA NLuc PTC, however, three Gly-tRNA$^{UGA}$ were identified with high suppression yields (~100-fold over background). Given the high sequence conservation among the Gly-tRNAs screened for anti-codon tolerance (Supplementary Figure 3), it would be difficult to predict de novo which tRNA would be most amenable to anticodon-editing.

**Identification of efficacious ACE-tRNA for PTC rescue.** We next performed screens on codon-edited tRNA for the each of the possible single nucleotide mutations which could produce a disease-causing PTCs: Arg-tRNA$^{UGA}$, Gln-tRNA$^{UAA}$, Gln-tRNA$^{UAG}$ Trp-tRNA$^{UGA}$, Trp-tRNA$^{UAG}$, Glu-tRNA$^{UAA}$, Glu-tRNA$^{UAG}$, Cys-tRNA$^{UGA}$, Tyr-tRNA$^{UAG}$, Tyr-tRNA$^{UAA}$, Leu-tRNA$^{UGA}$, Leu-tRNA$^{UAG}$, Leu-tRNA$^{UAA}$, Lys-tRNA$^{UAG}$, Lys-tRNA$^{UGA}$, Ser-tRNA$^{UGA}$, Ser-tRNA$^{UAG}$, and Ser-tRNA$^{UAA}$. The enzymatic activity of NLuc was not significantly influenced by the introduced amino acid (Supplementary Figure 4), therefore owing the difference in NLuc luminescence to ACE-tRNA suppression ability. The screen identified multiple ACE-tRNAs for each of the amino acids and stop codon type, with suppression coverage for all three stop codons (Fig. 2). Many of these ACE-tRNAs exhibited strong activity with >100-fold PTC suppression over background, which is significantly higher than the aminoglycosides used in this study (see below). Interestingly, some ACE-tRNAs displayed a clear preference for a particular anticodon editing, possibly reflecting altered aminoacyl-tRNA synthetase binding to the tRNA anticodon isoacceptor sequences[35]. For instance, tryptophan conversion to UAG suppression yielded rescue that was ten times higher than that of UGA editing of the same ACE-tRNA$^{Trp}$. Yet the opposite was true for glutamine, where a clear preference was shown for UAA over UAG. Notably, in each case, multiple high performing suppressors

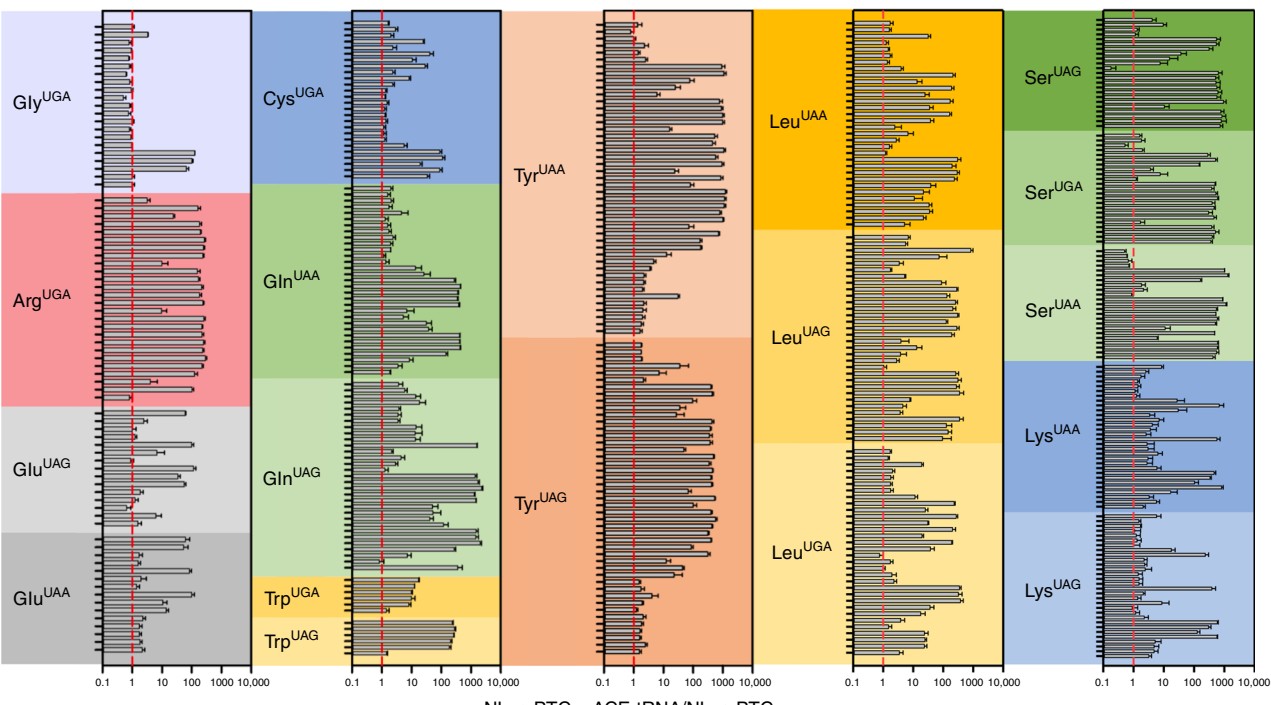

**Fig. 2** Screens of ACE-tRNA gene families with the high throughput cloning nonsense mutation reporter platform. The indicated anticodon edited PTC sequences were tested for each ACE-tRNA family that is one nucleotide away from the endogenous anticodon sequence, Supplementary Figure 1. Multiple high performing suppressor tRNA were identified for each class. Data are shown in Log10 scale in terms of normalized NLuc luminescence. Each tRNA dataset were obtained in triplicates and are displayed at average ± SEM. Coded identities and corresponding tRNA sequences are shown in Supplementary Figure 2a and Supplementary Data 1, respectively. Predicted cloverleaf structures for ACE-tRNAs for each family are included in Supplementary Figure 2b. The numerical values and ANOVA statistical analysis are located in Supplementary Data 2, where the order of tRNAs are maintained in top to bottom list form

were identified, and this was especially evident with $Arg^{UGA}$, a PTC which plays an outsized role in human disease; where twenty efficient ACE-$Arg^{UGA}$ suppressors were identified. In other cases, such as ACE-$tRNA^{Glu}$, of those which exhibited function, the suppression efficiency was roughly equal for UAA and UAG. And a similar pattern was found in ACE-$tRNA^{Lys}$, where encoding via UAG or UGA suppression were strongly mirrored. For Gln-$tRNA^{UAA}$, the suppression activity resulted in suppression signals >2000-fold over background. Of the ACE-tRNAs identified in the screen, the tryptophan tRNA gene family displayed the weakest suppression activity for UGA PTCs. With only six unique human ACE-$tRNA^{Trp}$ sequences available to screen, we sought to expand our UGA suppressing ACE-$tRNA^{Trp}$ library using tRNA from a range of species. We therefore tested UGA anticodon-editing tolerance for tryptophan tRNA genes with unique sequences from yeast, fly, mouse, rat, rabbit, and frog; in addition to a miscoding A9C $tRNA^{Trp}$ and bacterial Hirsh Trp suppressor[36–38], Supplementary Figure 5a. This effort was unsuccessful in identifying ACE-$tRNA^{Trp}$ UGA PTC suppression activity that exceeded that of the human ACE Trp tRNA, Supplementary Figure 5b. Overall, the tRNA screens identified multiple engineered tRNAs (for each amino acid and stop codon type) which displayed potent suppression, thus bearing general tolerance to anticodon editing.

**High-fidelity encoding and PTC suppression**. We next established whether ACE-tRNAs identified in our screen were functionalized at the expense of aminoacylation stringency by the cognate aminoacyl-tRNA synthetase. To this end, mass spectrometry was used to examine PTC suppression in a model soluble protein, histidinol dehydrogenase (HDH) (Fig. 3a). A TGA codon was introduced at asparagine 94 (N94) (Supplementary Figure 6) and co-expressed in HEK293 cells in tandem with plasmids encoding Glychr19.trna2 or Trpchr17.trna39 ACE-tRNAs, the top performing glycine and tryptophan ACE-$tRNA^{UGA}$,

respectively. The resulting full-length, suppressed, HDH proteins were purified via a Strep-Tactin® C-terminal affinity tag and analyzed by mass spectrometry, Fig. 3a (Supplementary Figure 6). Subsequent searches of the data identified the modification of Asn to Trp (+72 Da) for Trp chr17.trna39 and (−57 Da) for Glychr19.trna2, thus confirming the faithful encoding of the cognate amino acid for each ACE-tRNA type. Importantly, in each case >98% of the peptide identified at the HDH p.N94X site had the encoded cognate tryptophan and glycine. Further, both ACE-tRNAs retained selectivity for the UGA stop codon, over UAA and UAG, Fig. 3b (ACE-$tRNA^{Gly}$) and Supplementary Figure 7 (ACE-$tRNA^{Trp}$). Lastly, when transiently expressed, the ACE-$tRNA^{Gly}$ outperformed the conventional small molecule suppressors gentamicin (40 µM) and G418 (140 µM) in their ability to suppress NLuc-UGA stably expressed in HEK293 cells (Fig. 3c). The same was true even for ACE-$tRNA^{Trp}$, which had a lower suppression efficiency yet exceeded PTC rescue compared to G418, Supplementary Figure 8.

**Minimal suppression activity at protein termination codons**. We next raised the question of whether ACE-tRNAs that show efficacious suppression of premature stop codons may also induce global readthrough of native stop codons. To address this potential "off target" suppression, a transcriptome-wide quantitative profile of actively engaged ribosomes on all cellular transcripts was obtained by generating libraries of ribosome footprints from HEK293 cells expressing exogenous ACE-tRNAs or a control mock plasmid (puc57GG). Streptomycin was removed from the growth media to prevent readthrough artifacts. For comparison, we also generated the ribosome footprint library from cells in the presence or absence of G418 (150 µM, 48 h). Figure 4a shows ribosome footprint densities of G418 and five ACE-tRNAs compared against controls (log2-fold change) on 3′ UTR regions. Only transcripts with a minimum threshold of 5 RPKM in the coding sequence and 0.5 RPKM in the 3′UTR in two replicate libraries were included for the quantitation

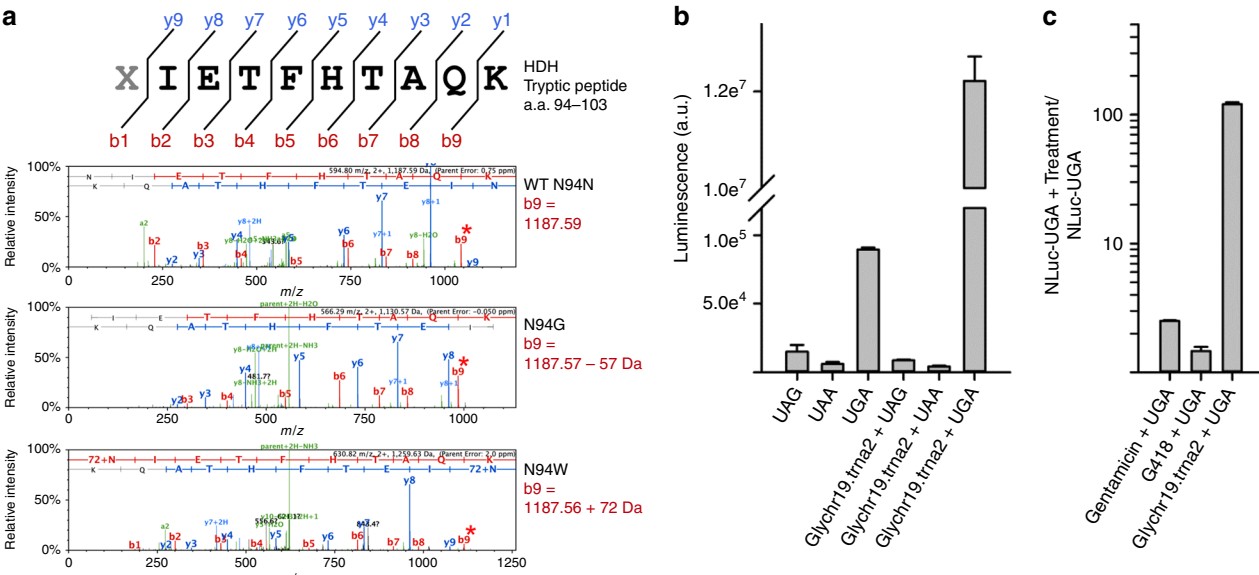

**Fig. 3** Cognate encoding and high-fidelity suppression by engineered tRNA. **a** Tryptic fragment of histidinol dehydrogenase (HDH), where X indicates suppressed PTC codon. MS/MS spectra of the tryptic fragment with masses of indicated y and b ions for WT (top), N94G (middle), and N94W (bottom) HDH. b9 ion mass is shifted by the predicted mass of −57 Da and +72 Da from the WT asparagine, indicating the encoding of cognate amino acids glycine and tryptophan by ACE-$tRNA^{Gly}$ and ACE-$tRNA^{Trp}$ (Trp-chr17.trna39), respectively. **b** ACE-TGA - $tRNA^{Gly}$ (Glychr19.trna2) selectively suppresses the UGA stop codon in transiently transfected HEK293 cells. **c** ACE-$tRNA^{Gly}$ transfection outperforms both gentamicin (40 µM; n = 3) and G418 (140 µM; n = 3) following a 48 h incubation in Hek293 cells stably expressing NLuc-UGA. Data is presented as standard error of mean in **b** and **c**

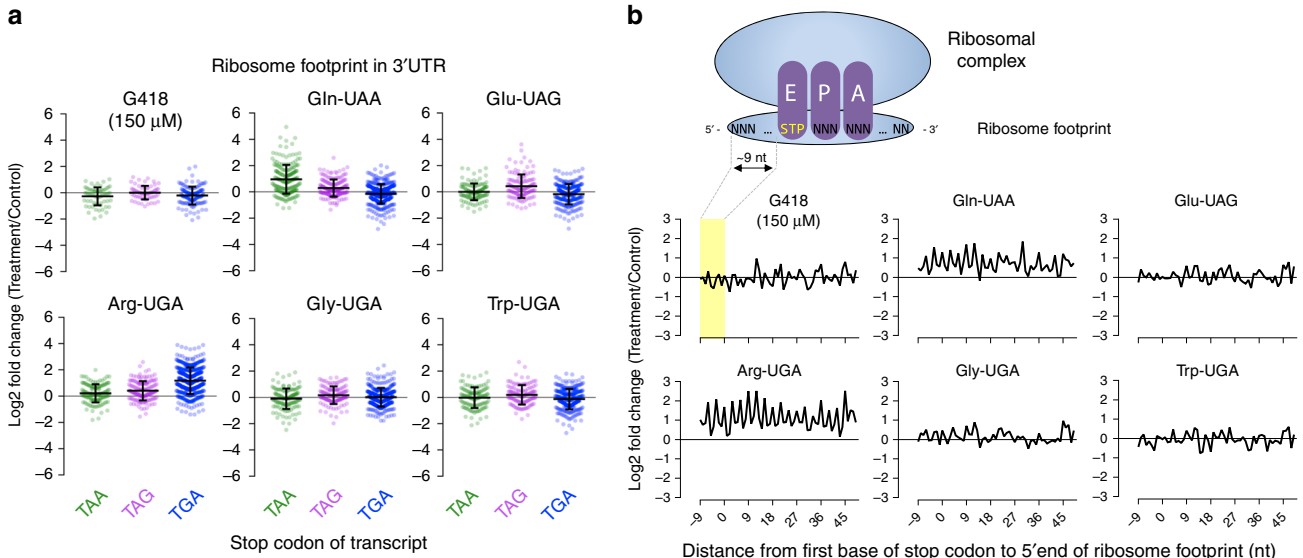

**Fig. 4** Ribosome profiling of ACE-tRNA on transcriptome-wide 3′UTRs. **a** Ribosome footprint densities on 3′UTRs are plotted as log2-fold change for reads of treated cells vs. control (puc57GG empty vector) as described in the materials and methods. Transcripts were grouped by their endogenous UAA, UAG, and UGA stop codons. Each point represents the mean of two replicates for a transcript. Error bars show Mean ± SD of the log2-fold changes. **b** The average log2-fold change of normalized ribosome footprint occupancy was plotted for each nucleotide from −9 to +50 nt surrounding stop codons of transcriptome (18,101 sequences). The cartoon illustrates the ~9 nt offset from the 5′ end of ribosome footprint to the first base position of stop codon in the ribosome E-site. ACE-tRNA genes used for 4× constructs are Trp-chr17.trna39 (UGA), Gly-chr19.trna2 (UGA), Arg-chr9.trna6 (UGA), Gln-chr17.tRNA14 (UAA), and Glu-chr13.trna2 (UAG). Sequences are located in Supplementary Data 1

comparison (254 transcripts in G418 and 495–748 transcripts in ACE-tRNAs). In this system, G418 had no observable effect on transcriptome-wide 3′UTR ribosome density for any of the three endogenous stop codon groups. ACE-tRNAs examined here had no detectable change of 3′UTR ribosome density with the exception of ACE-tRNA Gln-UAA and Arg-UGA which induced approximately a 2-fold increase in 3′UTR ribosome density for the cognate stop codon complimentary to the ACE-tRNA anticodon. Understanding the biological significance of 2-fold readthrough of protein stops will require further study, but this effect is substantially lower compared to the 100-fold to 1000-fold suppression of PTC for the same ACE-tRNA.

Multiple in-frame stop codons are frequently found at the end of genes[39–41] and may cause a minor difference in overall 3′UTR ribosome density for ACE-tRNA and G418 treatment. We therefore examined ribosome occupancy at each nucleotide in the 3′UTR within a 60 nt region downstream of the stop codons. Figure 4b demonstrates the ribosome occupancy surrounding native stop codons in the 3′ UTR for each nucleotide within the region from +6 to +65 nt relative to the first nucleotide of stop codon. Reads were normalized per total million-mapped reads, compared against control cells, and reported as a log2-fold change as in panel a. More than 2313 transcripts were mapped to at least 1 footprint in the region of interest. ACE-tRNA Gln-UAA and Arg-UGA showed not only notable increased ribosome occupancy in the early region but also characteristic 3-nt periodicity, indicating that the ribosomes were not randomly distributed but followed codon-by-codon movement. ACE-tRNAs for Trp-UGA, Gly-UGA and Glu-UAG, or G418, consistently showed no observable change of ribosome occupancy even in the early region of 3′UTR. Taken together, the ribosome profiling data argue that efficiency of native stop codon suppression by ACE-tRNAs is generally low, and markedly less than the level of PTC suppression.

**Potent in vivo and in vitro suppression activity.** We next examined the in vivo activity and stability of ACE-tRNA. We

delivered the NLuc-UGA PTC reporter cDNA together with a plasmid encoding four copies of the ACE-tRNA^Arg UGA or an "empty vector control" into mouse skeletal muscle (tibialis anterior) using electroporation[42–44]. We then compared these data to the expression of the WT NLuc. The results showed that the ACE-tRNA^Arg UGA is a potent in vivo PTC suppressor, yielding expression profiles equal to or at some time points, greater than, the full-length WT NLuc (Fig. 5a). The signal from the NLuc-UGA plasmid and non-electroporated muscle was undetectable. Further, ACE-tRNA^Arg suppression activity was stable, as evidenced by the similar duration of NLuc activity between rescued and WT protein (Fig. 5b). Furthermore, this duration and intensity of luciferase expression is generally supportive of in vivo tolerability with ACE-tRNA^Arg. However, additional experimentation will be needed to acquire delivery dependent expression and to identify any potential tissue specific tolerance issues. We next wanted to determine if functional ACE-tRNAs can be delivered as RNA. To this end, we transfected ACE-tRNA^Trp and ACE-tRNA^Gly RNA transcripts into HEK293 cells that stably express the NLuc-UGA reporter. Here the results indicated that both ACE-tRNAs functioned similarly as when expressed as cDNA plasmids, with comparable fold rescue when delivered as small RNA (Fig. 5c). In separate experiments, we compared the duration of ACE-tRNA rescue delivered as RNA and cDNA to HEK293 cells (Supplementary Fig. 11). ACE-tRNA activity delivered as RNA peaked at 8 h following delivery and decreased over a 48 h period, whereas cDNA expression plasmids supported an increase in PTC suppression activity that plateaus at 48 h. Next, we sought to rescue two disease causing mutations in cystic fibrosis transmembrane conductance regulator (CFTR). This large membrane protein controls anion transport across epithelia in multiple organs and missense and nonsense mutations within its reading frame cause cystic fibrosis. To this end, CFTR p.G542X (c.16524G>T; TGA stop codon) and p.W1282X (c.3846G>A; TGA stop codon) cDNAs were transiently coexpressed with their respective ACE-tRNA expression plasmids

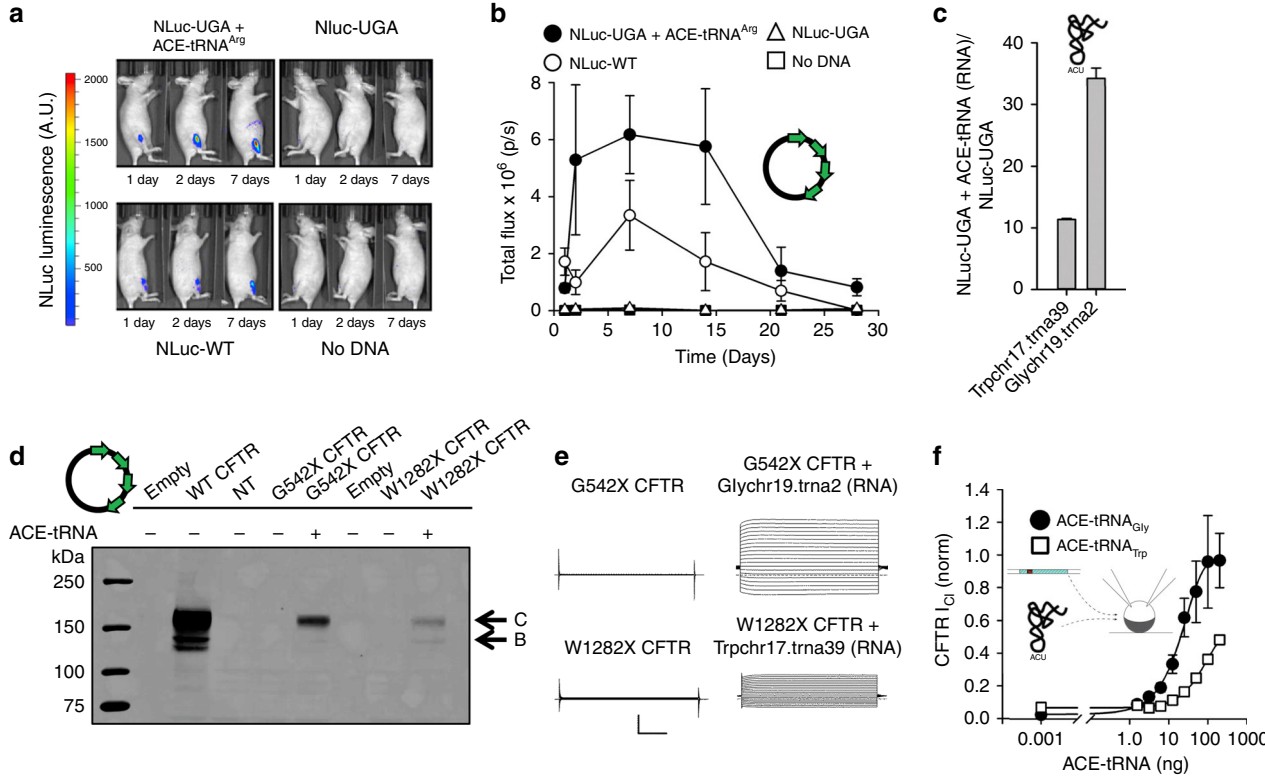

**Fig. 5** In vivo delivery and suppression with ACE-tRNA as cDNA and RNA. **a** Representative images of mice injected with NLuc-UGA with ACE-tRNA[Arg] (Arg-chr9.trna6 UGA) or pUC57 empty vector, NLuc-WT or water in the tibialis anterior muscle followed by electroporation at days 1, 2, and 7 after DNA administration. **b** Quantification of luminescence emission by the tibialis anterior muscles of the abovementioned mouse groups at different timepoints after DNA injection and electroporation ($n = 3$ mice per group, data are shown as SEM). **c** Rescued luminesce of stably expressed NLuc–UGA following transfection of Trpchr17.trna39cRNA and Glychr19.trna2 RNA transcripts ($n = 3$ for each ACE-tRNA). **d** Representative western blot analysis of CFTR protein expressed in HEK293 cells 36 h following transfection of WT, G542X, G542X + Glychr19.trna2, W1282X and W1282X + Trpchr17.trna39 CFTR cDNA. **e** Exemplar families of CFTR Cl⁻ current traces recorded using two-electrode voltage-clamp, 36 h following injection with WT, G542X, G542X + ACE-tRNA-Glychr19.trna2, W1282X and W1282X + ACE-tRNA-Trpchr17.trna39 CFTR cRNA. Currents were elicited using 5 mV voltage steps from −60 to +35 mV. The vertical and horizontal scale bars indicate 10 μA and 50 ms, respectively. **f** Dose response of G542X ACE-tRNA[Gly] (filled circles; $n = 11$–19) and W1282X ACE-tRNA[Trp] (open squares; $n = 17$–22) rescue (CFTR Cl⁻ currents elicited at +35 mV were normalized to WT CFTR Cl⁻ currents at +35 mV). ACE-tRNA[Gly] rescue achieves WT-level of expressed CFTR current. Data is presented as standard error of mean for panels **c** and **f**

in HEK293 cells and analyzed by western blot using a C-terminal antibody to identify production of the full-length protein (Fig. 5d). Both rescue conditions, as well as WT CFTR expression, resulted in successfully trafficked CFTR protein as evidence by the presence of both the fully glycosylated band C form and the core glycosylated band B CFTR protein. No signal was seen for either p.G542X or p.W1282X transfected alone, indicating a low rate of spontaneous read-through of the indicated PTC under these conditions. To better quantify the PTC suppression properties of each ACE-tRNA in the absence of delivery or expression caveats, we turned to the *Xenopus leavis* oocyte, a non-dividing model cell in which the ACE-tRNA concentration (as RNA) can be controlled and functional expression can be quantitated. Specifically, this expression system is amenable to microinjection and two-electrode voltage-clamp (TEVC) analysis, a facile electrophysiological method for assessing ion channel function at the plasma membrane. CFTR cRNA (complementary RNA produced in vitro from a cDNA template) was injected alone or together with the indicated ACE-tRNA RNA at increasing concentrations (Fig. 5e, f). Functional CFTR channels were not seen for either mutant lacking co-injected ACE-tRNA, even in the presence of a maximal CFTR activation cocktail, forskolin (10 μM; adenylate cyclase activator) and 3-isobutyl-1-methylxanthine (1 mM; phosphodiesterase inhibitor), (Fig. 5e, left). However, under the

same conditions, when co-injected with 200 ng of ACE-tRNA Gly chr19.trna2 (Fig. 5e, top right) or Trp chr17.trna39 (Fig. 5e, bottom right) CFTR chloride conductance was measured in response to transient changes in membrane potential, indicating that both ACE-tRNAs were highly efficacious at suppressing two disease-causing UGA PTCs. To better quantify the relative expression of rescued channels, we compared this rescue to WT CFTR cRNA alone (25 ng), and assessed suppression of PTCs in CFTR across a range of ACE-tRNA concentrations. The resulting ACE-tRNA dose response "current-voltage" relationships are shown in Fig. 5f. These data were generated by plotting the steady state ionic current at each voltage vs. the voltage used to elicit the measured currents and are a direct measure of channel function and abundance. WT-like current levels of expression were achieved by Gly chr19.trna2, and ~50% for Trp chr17.trna39 ACE-tRNAs, consistent with the predetermined suppression activity and cognate amino acid encoding for these tRNA. When rescued CFTR currents were normalized to WT currents at +35 mV, it can be observed that ACE-tRNA[Gly] (black circles) PTC suppression saturates at 100 ng while ACE-tRNA[Trp] (white squares) does not (Fig. 5f). Through this analysis, we can estimate that ACE-tRNA[Trp] RNA transcripts ($EC_{50} \cong 3.9 \mu M$) are less efficacious than ACE-tRNA[Gly] ($EC_{50} \cong 838 nM$) at suppressing their respective CFTR nonsense mutations.

## Discussion

PTCs cause a multitude of human diseases and there are no established therapeutic options for their therapeutic management. Herein, we report the high-throughput cloning and identification, characterization and functional analysis of anticodon-edited tRNA which display efficacious PTC reversion in eukaryotic cells and mouse skeletal muscle. Notably, our screen identifies ACE-tRNA, in total, with the potential to repair a vast majority of known human disease-causing PTC, but this therapeutic will require overcoming tissue and delivery specific challenges. However, the engineered tRNA, once delivered, faithfully encode their cognate amino acid, thus abrogating spurious effects on downstream protein stability, folding, and trafficking, and consequently negating the need for tandem therapies involving protein folding or trafficking agents. When transfected as cDNA, ACE-tRNAs rescued multiple full-length proteins via PTC suppression; a NLuc luciferase reporter, a model protein HDH, and two disease nonsense mutations in *CFTR*. Potent and stable in vivo PTC suppression in mouse skeletal muscle was displayed by an ACE-tRNA[Arg] cDNA, suggesting a particularly high level of cellular tolerance for ACE-tRNA activity. The identification of an active ACE-tRNA for arginine in muscle is relevant for the treatment of dystrophinopathies caused by nonsense mutations. Following suit with most genetic diseases, greater than 10 percent of dystrophinopathies are caused by nonsense mutations[45], where CGA->TGA mutations are most prevalent[45]. Efficient suppression was also achieved with ACE-tRNAs delivered as synthetic RNA transcripts, thus enabling the development of nanoparticle formulations. Future studies will be needed to assess ideal tRNA delivery strategies for each tissue and disease type, where efforts will likely benefit from rapidly expanding technologies for nucleic acid delivery. One of the factors that has hampered PTC suppression therapeutics is the dearth of sufficient nonsense-associated animal models of disease and/or PTC suppression reporter animals. Using the rapid and precise CRISPR/Cas mouse genome manipulation methodologies, generation of appropriate nonsense-associated animal models of disease will greatly facilitate advances in PTC therapeutics centered around traditional small molecules (i.e., aminoglycoside chemistries), nuclease technologies (i.e., CRISPR/Cas and TALENS) and ACE-tRNA based approaches described here.

Agents that suppress PTCs have the potential to also produce readthrough of native stop codons. The RNA profiling data presented herein suggest this is generally not the case in cells for the codon-edited tRNA that we tested. While detectable readthrough was found with Arg-tRNA[UGA] and Gln-tRNA[UAA], no significant effect on global translation termination was measured with Glu-tRNA[UAG], UGA-Gly-tRNA[UGA], and Trp-tRNA[UGA]. This behavior did not obviously segregate with stop codon type, or the intrinsic PTC suppression activity of the tRNA. One potential mechanism that results in ACE-tRNA unable to promote readthrough at real stop codons may be the contextual sequence landscapes near translation terminations[46]. This possibility is supported by the finding that the composition of termination complexes at PTCs differ from those at native stops[47,48]. However, in cases where lower level readthrough occurs, there are multiple cellular mechanisms in place to limit both normal stop read-through and damaging effects thereof. Multiple in-frame stop codons are frequently found at the end of genes[39–41] and specialized ubiquitin ligases[49] and ribosome associated pathways[50] are known to identify and degrade proteins with erroneous translation termination. Notably, selenocysteine is encoded at the UGA codon within a compulsory selenocysteine insertion sequence (SECIS) where incorporation is tightly regulated by a complex of specialized proteins, such as the tRNA[Sec] translation elongation factor (eEFSec)[51–53]. Nonetheless, despite the

relatively limited impact on protein termination sites depicted in the profiling data in mammalian cells exposed to transient expression, similar ribosomal profiling experiments should be performed in the desired cell, tissue type, and codon site for ACE-tRNA delivery and expression.

Previous studies have shown that the surrounding mRNA sequence influences inherent stop codon suppression efficacy of aminoglycosides and Ataluren PTC[54–57], and ACE-tRNA may be similarly affected. Further, while gene addition strategies to replace a PTC containing gene, via viral or non-viral delivery, have achieved short term benefit in some settings, it may be difficult to regulate transgene expression levels. In contrast, the abundance of protein rescue via ACE-tRNA suppression is coupled to native cellular RNA levels, and thus upper levels of expression will be intrinsically regulated. The biological purpose remains unknown for a majority of the variable isoacceptor tRNA sequences in the human genome, and almost half these genes have been speculated to be transcriptionally silent pseudogenes[58], however the data here suggest many annotated tRNA are viable. Consistent with this possibility, a suppression approach has been used to identify functional isodecoder tRNAs within Ser and Leu isoacceptor families[59]. The data we present here further demonstrate that the majority of tRNA gene sequences support viable activity when removed from the genomic context, further deepening the mystery for the biological need for a plurality of tRNA, tissue specific expression, and codon usage. Thus, the high-throughput suppression strategy described here will be useful to identify new types of tRNA sequences with unique suppression properties, and such studies have the potential to produce new RNA reagents as well as advance the molecular understanding tRNA expression and suppression.

## Methods

**Nonsense reporter HTC plasmid.** The parent plasmid used was pcDNA3.1(+). The cDNA encoding pNLuc was Gibson Assembled (New England Biolabs, USA) into restriction sites HindIII and XhoI. A glycine (codon gga), tryptophan (tgc), amber (tag), opal (tga) and ochre (taa), were added to amino acid position 160 during cDNA pcr. The pcDNA3.1(+) polyA sequence was replaced for one with no BbsI restriction sites using pcr based Gibson Assembly. The high throughput ACE-tRNA Golden Gate cloning site was generated by first inserting the 5′ leader sequence of the human tRNA[Tyr] gene (bold) with a T7 promoter sequence upstream (italics) (*TAATACGACTCACTATAG* **AGCGCTCCGGTTTTTC TGTGCTGAACCTCAGGGGACGCCGACACACGTACACGTC**)[60] followed by two BbsI restriction sites (underlined) (TAGTCTTCGG (*ccdB cassette*) AAGAA-GACCG) and 3′ termination sequence (bold) followed by a reverse T3 primer sequence (italics) (**GTCCTTTTTTTG***CTTTAGTGAGGGTTA ATT*). See Supplementary Table 2 for sequences of oligos used for cloning as pNanoLuc-TGA pcDNA3.1(+) sequence.

**HTC of ACE-tRNA library.** tRNA gene sequences were obtained from the tRNA database tRNAscan-SE (http://gtrnadb.ucsc.edu/index.html; PMID: 26673694). Sequences of all tRNA genes used in this study are numbered in Supplementary Figure 2a and Supplementary Data 1. tRNA sequences were synthesized as complementary Ultramers from Integrated DNA Technologies (IDT, USA) in 96-well format at 200 pmol scale with their corresponding anticodons mutated appropriately (UAG, UGA, or UAA). All tRNA sequences were synthesized with CGTC and GGAC overhangs (annotated 5′->3′) on forward and reverse oligos, respectively. Ultramers were annealed by resuspending in annealing buffer (100 mM Potassium Acetate; 30 mM HEPES, pH 7.5) to 100 ng/µl, heated to 96 °C for 2 min and cooled at 1 °C/min in a thermocyler to 4 °C. In 96-well PCR plates, each well contained 10 ng of HTC plasmid with appropriate PTC codon, 2 ng ACE-tRNA duplex, 1 mM ATP, 10 mM DTT, 400 units T4 DNA Ligase, 1µl 10x CutSmart® Buffer (New England Biolabs, USA) and 10 units BbsI-HF, queued to 10 µl with ddH$_2$O. The 96 well plates were cycled as follows ([5 min @37 °C, 5 min @20 °C] × 30 cycles, 10 min @ 37 °C, 10 min @ 80 °C and cooled to 4 °C in a thermocycler. In a deep welled 96-well plate 1 µl of the Golden Gate reaction was added to 10 µl of DH5α chemically competent cells (ThermoFisher, USA), heat-shocked @ 42 °C for 30 s and resuspended in 100 µl of Super Optimal Broth (S.O.C.; Thermofisher, USA). Transformations were outgrown at 37 °C for 1 h, 250 rpm and then added to 2 ml of Luria-Bertani liquid media (LB) supplemented with 100 µg/ml Carbenicillin and grown in covered deep 48-well plates @ 37 °C for 20 h, 300 rpm. E. coli out-growth was performed in deep well plates and clamps from Enzyscreen (http://www.enzyscreen.com). E. coli suspension cultures were spun down (10 min,

4000×*g* at RT) and plasmid DNA was prepared and diluted to 125 ng/μl (IBI scientific, USA). All clones were sequence verified. Using this method, we achieved 100% cloning efficiency.

**HTS of ACE-tRNA library.** The day before transfection, HEK293 cells (ATCC, USA) (<40 passages) were plated at $1.4 \times 10^4$ cells/well in 96-well cell culture treated plates in Dulbecco's Modified Essential Medium (DMEM) supplemented with 10% FBS, 1% Pen/Step and 2 mM L-Glutamine (Thermofisher, USA). The all-in-one nonsense reporter with ACE-tRNA genes were transfected in triplicate/plate using Calfectin (Signagen, USA). Sixteen hours post-transfection, the media was aspirated and 20 μl of PBS was added to each well. Fifteen microliter of lytic Nano-Glo® Luciferase Assay Reagent was added to each well (1:50 reagent to buffer; Promega, USA). The plates were incubated for 2 min after rotational shaking and read using a SpectraMax i3 plate reader (Molecular Devices, USA; integration time, 200 ms; All wavelengths collected in endpoint mode). Luminescence was averaged across three wells for each experiment and all ACE-tRNAs were repeated >3 times in this fashion. Each plate also contained in triplicate wells transfected with the all-in-one nonsense reporter with no ACE-tRNA to server as control for transfection efficiency and baseline PTC readthrough. All values are reported as ratios of ACE-tRNA luminescence over baseline PTC readthrough luminescence ± SEM. One-way ANOVAs were performed with Tukey's post-hoc analysis across all ACE-tRNAs in a given amino acid family, Supplementary Data 2.

**Expression plasmids.** For expression in mammalian cells, the cDNA for the coding region and 200 base-pair of the 3′ untranslated region (UTR) of human CFTR was ligated into pcDNA3.1(+) (Promega, USA) using the KpnI and XbaI restriction enzymes. The G542tga and W1282tga mutations were introduced by BioBasic Inc. (Canada). For expression in *Xenopus laevis* oocytes, the cDNA for the coding region and 140 base-pair of the 5′ and 244 base-pair 3′ UTR of human CFTR was ligated into pGEM-HE (Promega, USA). The cDNA encoding the *E. coli* histidinol dehydrogenase was codon optimized for *Mus musculus* and synthesized (BioBasic Inc, Canada) with a c-terminal 8xHis-Strep-tag for protein purification from mammalian cells (Supplementary Figure 6). The synthesized cDNA was ligated into pcDNA3.1(+) using EcoRI and XhoI restriction sites. The nonsense mutation TGA was introduced by BioBasic Inc. To generate multiplexed ACE-tRNA expression plasmids, we generated a novel parent Golden Gate pUC57(amp) plasmid by inserting a BbsI "multiple cloning site" (5′-GAATTCTTCCCGAGA CG**TTCC**AAGTCTTCATG**AAGACTA**CA**GG**CGTCTCCCAGGAAGCT-3′; directional BbsI recognition sequences are underlined and unique four base-pair overhangs for ligation are bolded) between the EcoRI and HindIII restriction sites. pUC57 (amp) was chosen as a parent plasmid because it is relatively small in size and lacks backbone BbsI restriction sites and T7 and T3 promoter sequence. A feature included in the HTS plasmid is T7 and T3 promoter sequence flanking the ACE-tRNA cassette, giving universal primer binding sequences with comparable melting temperatures ($T_m$), ideal for pcr amplification. Using the NEB Golden Gate Assembly Tool (https://goldengate.neb.com/editor) we generated pcr primers that annealed to the T7 and T3 flanking sequence and created unique four base-pair overhangs following cleavage of distal BbsI recognition sequence. The end result was the generation of four ACE-tRNA pcr products using universal pcr primers that could be daisy-chained through complementary four base-pair overhangs and ligated into the puc57 Golden Gate plasmid using a one-pot Golden Gate reaction. All clones were sequence verified. ACE-tRNA genes used for 4x constructs in Figs. 4 and 5 are Trp-chr17.trna39 (UGA), Gly-chr19.trna2 (UGA), Arg-chr9.trna6 (UGA), Gln-chr17.tRNA14 (UAA) and Glu-chr13.trna2 (UAG). ACE-tRNA sequences are located in Supplementary Data 1 and cloning oligos are listed in Supplementary Table 2.

**Protein expression and western blot.** HEK293T cells (ATCC, USA) were grown in standard grown media containing (% in v/v) 10% FBS (HiClone, USA), 1% Pen Strep, 1 % L-Glut in high glucose DMEM (Gibco, USA) at 37 °C, 5% CO₂. cDNA was transfected at 75% confluency using Calfectin according to standard protocols (SignaGen Laboratories, USA). Following 36 h the cells were scraped and pelleted at 7000×*g* for 8 min at 4 °C in PBS supplemented with 0.5 μg/ml pepstatin, 2.5 μg/ml aprotinin, 2.5 μg/ml leupeptin, 0.1 mM PMSF, 0.75 mM benzamidine. For CFTR expressing cells, the cell pellet was vigorously dounced in 100 mM sucrose, 150 mM NaCl, 1 mM DTT, 0.5 μg/ml pepstatin, 2.5 μg/ml aprotinin, 2.5 μg/ml leupeptin, 0.1 mM PMSF, 0.75 mM benzamidine, 50 mM Tris-HCL ph 7.4 and centrifuged at 100,000×*g* to separate total membranes from the soluble cytosolic proteins. Pellets were solubilized in a buffer containing 1% triton, 250 mM NaCl, 50 mM tris-HCl pH 7.4, and 0.5 μg/ml pepstatin, 2.5 μg/ml aprotinin, 2.5 μg/ml leupeptin, 0.1 mM PMSF, 0.75 mM benzamidine. Equal cell-lysate was loaded on a 3–15% separating gradient SDS-page with 4% stacking gel in the presence of 1% 2-mercaptoethanol, separated at 55 V O/N and transferred to 0.45 μM LF PVDF (Bio-Rad, USA). PVDF was immunoblotted using anti-CFTR antibody (1:1000; M3A7, Millipore, USA) in 2% non-fat milk and imaged on LI-COR Odyssey Imaging System (LI-COR, USA). For HDH-His-Strep expressing cells, the cell pellet was vigorously dounce homogenized in 100 mM sucrose, 1 mM DTT, 1 mM EDTA, 20 mM tris-HCl pH 8.0, 0.5 μg/ml pepstatin, 2.5 μg/ml aprotinin, 2.5 μg/ml leupeptin, 0.1 mM PMSF and 0.75 mM benzamidine. The lysate was centrifuged at

100,000×*g* for 30 min at 4 °C. The supernatant (soluble cellular protein) was separated on 4–12% Bis-Tris SDS-page acrylamide gels (ThermoFisher, USA) in the presence of 1% 2-mercaptoethanol, transferred to 0.22 μM LF PVDF (Bio-Rad, USA) and immunoblotted using anti-Strep antibody (1:5000; iba, Germany) in 2% non-fat milk and imaged on LI-COR Odyssey Imaging System (LI-COR, USA).

**Mass spectrometry.** Fragmentation data on purified HDH-His-Strep protein were obtained at the University of Iowa Proteomics Facility. Briefly, HDH-His-Strep protein from the soluble fraction of the high-speed spin was passed through StrepTrap HP columns (GE Healthcare, Sweden) and washed with 5 column volumes of 100 mM sucrose, 1 mM DTT, 1 mM EDTA, 20 mM tris-HCl pH 8.0, 0.5 μg/ml pepstatin, 2.5 μg/ml aprotinin, 2.5 μg/ml leupeptin, 0.1 mM PMSF, and 0.75 mM benzamidine. The protein was eluted in wash buffer supplemented with 10 mM d-desthbiotin and concentrated in 30kDA cutoff Amicon-Ultra filtration columns (Millipore, USA). The concentrated protein was loaded on NuPage 4–12% Bis-Tris precast gels (Invitrogen, USA) and separated at 150 V for 1.5 h. The gel was stained using a Pierce mass spectrometry compatible silver stain kit (ThermoFisher Scientific, USA).

**In-gel trypsin digestion.** Briefly, the targeted protein bands from SDS-PAGE gel were manually excised, cut into 1 mm³ pieces, and washed in 100 mM ammonium bicarbonate:acetonitrile (1:1, v/v) and 25 mM ammonium bicarbonate /acetonitrile (1:1, v/v), respectively to achieve complete destaining. The gel pieces were further treated with ACN, and dried via speed vac. After drying, gel pieces were reduced in 50 μl of 10 mM DTT at 56 °C for 60 min and then alkylated by 55 mM IAM for 30 min at room temperature. The gel pieces were washed with 25 mM ammonium bicarbonate:acetonitrile (1:1, v/v) twice to removed excess DTT and IAM. After drying, the gel pieces were placed on ice in 50 μl of trypsin solution at 10 ng/μl in 25 mM ammonium bicarbonate and incubated on ice for 60 min. Then, digestion was performed at 37 °C for 16 h. Peptide extraction was performed twice for 0.5 h with 100 μl 50% acetonitrile/0.2% formic acid. The combined extracts were concentrated in a Speed Vac to ~15 μl.

**LC-MS/MS.** Our mass spectrometry data were collected using an Orbitrap Fusion Lumos mass spectrometer (Thermo Fisher Scientific, San Jose, CA) coupled to an Eksigent Ekspert™ nanoLC 425 System (Sciex). A Trap-Elute Jumper Chip (P/N:800–00389) and a coupled to a 1/16" 10 port Valco directed loading performed by the gradient 1 pump and final elution (by the gradient 2 pump). The column assembly was was designed as two tandem 75 μm × 15 cm columns (ChromXP C18-CL, 3 μm 120 A, Eksigent part of AB SCIEX) mounted in the ekspert™ cHiPLC system. For each injection, we loaded an estimated 0.5 μg of total digest. Peptides were separated in-line with the mass spectrometer using a 120 min gradient composed of linear and static segments wherein Buffer A is 0.1% formic acid and B is 95% ACN, 0.1% Formic acid. The gradient begins first holds at 4% for 3 min then makes the following transitions (%B, min): (26, 48), (35, 58), (35, 64), (50, 72), (50, 78), (94, 84), (94, 96), (4, 100), (4, 120).

**Tandem mass spectrometry on the LUMOS Orbitrap.** Scan sequences began with a full survey (*m/z* 350–1500) acquired on an Orbitrap Fusion Lumos mass spectrometer (Thermo) at a resolution of 60,000 in the off axis Orbitrap segment (MS1). Every 3 s of the gradient MS1 scans were acquired during the 120 min gradient described above. The most abundant precursors were selected among 2–8 charge state ions at a 2.0E5 threshold. Ions were dynamically excluded for 30 s if they were targeted twice in the prior 30 s. Selected ions were isolated by a multi-segment quadrupole with a mass window on *m/z* 2, then sequentially subjected to both CID and HCD activation conditions in the IT and the ioin routing multipole respectively. The AGC target for CID was 4.0E04, 35% collision energy, an activation Q of 0.25 and a 100 ms maximum fill time. Targeted precursors were also fragmented by high energy collision-induced dissociation (HCD) at 40% collision energy, and an activation Q of 0.25. HCD fragment ions were analyzed using the Orbitrap (AGC 1.2E05, maximum injection time 110 ms, and resolution set to 30,000 at 400 Th). Both MS2 channels were recorded as centroid and the MS1 survey scans were recorded in profile mode.

**Proteomic searches.** Initial spectral searches were performed with Proteome Discoverer version 2.1.1.21 (ThermoFisher Scientific, USA) using Sequest HT. Spectra were also searched with Byonic search engine (Protein Metrics) ver. 2.8.2. Search databases were composed of the Uniprot KB for species 9606 (Human) downloaded 10/24/2016 containing 92,645 sequences and Uniprot KB for taxonomy 562 (*E. coli*) downloaded on 11/08/2016 containing 10,079 sequences. For Byonic searches, these two data bases were directly concatenated. In either search an equal number of decoy entries were created and searched simultaneously by reversing the original entries in the Target databases.

**In vitro cRNA transcription.** G542X$_{UGA}$, W1282X$_{UGA}$, and WT CFTR pGEMHE (Mense et al., 2006; PMID:1703051) plasmids were linearized by 10× excess of NheI-HF restriction enzyme (site positioned 3′ of coding region)(New England BioLabs, USA) for 3 h at 37 °C and purified using standard cDNA precipitation

methods. All cRNAs were transcribed using the mMessage mMachine T7 Ultra Kit (ThermoFisher Scientific, USA). Purification of the cRNA from the transcription reaction was conducted on columns from the RNeasy Mini Kit (Qiagen, Germany). Concentration was determined by absorbance measurements at 260 nm and quality was confirmed on a 1% agarose gel (RNase-free). All cRNA was queued to 1μg/ml before use and all results were generated from ≥2 cRNA preparations.

**In vitro tRNA transcription**. Trpchr17.trna39 and Glychr19.trna2, the top performing Trp and Gly ACE-tRNAs, were transcribed in vitro using CellScript T7-Scribe Standard RNA IVT Kit (CELLSCRIPT, USA). Equimolar concentration of T7 oligo (5′-taatacgactcactata-3′) was annealed to ACE-tRNA PAGE-purified Ultramers (20ug; Integrated DNA Technologies, Coralville, IA) coding for the ACE-tRNA and preceded by a T7 promoter (italics). Importantly, the three terminal nucleotides containing CCA were included (bold).

Trpchr17.trna39 (3′->5′):
**TGG**TGACCCCGACGTGATTTGAACACGCAACCTTCTGATCTGAAGTCAG ACGCGCTACCGTTGCGCCACGAGGCC*TATAGTGAGTCGTATTA*

Glychr19.trna2 (3′->5′):
**TGG**TGCGTTGGCCGGGAATCGAACCCGGGTCAATGCTTTGAAGGAGCTA TGCTAACCATATACCACCAACGC*TATAGTGAGTCGTATTA*

The total reaction volume was adjusted to 100 μl and the kit reagents were added in the following amounts: 10 μl of 10× T7-Scribe transcription buffer, 7.5 μl of each nucleotide (100 mM stocks), 10 μl of 100 mM Dithiothreitol, 2.5 μl ScriptGuard RNase Inhibitor, 10 μl T7-Scribe enzyme solution. After the reaction was incubated for 4–5 h at 37 °C, the DNA template was digested with 5 μl DNase (1 U/μl) provided with the kit for 30–60 min. The ACE-tRNA was extracted from the reaction with acidic phenol chloroform (5:1, pH 4.5) and precipitated with ethanol. The precipitated ACE-tRNA was pelleted, washed, dried and resuspended in 100 μl DEPC-treated water and further purified with Chroma Spin-30 columns (Clontech, USA). The procedure yielded roughly 100 μl of ~5 μg/μl ACE-tRNA. ACE-tRNAs were re-pelleted in 20 μg aliquots, washed, lyophilized and stored at −80 °C until use. All results were generated from ≥2 ACE-tRNA preparations.

**Ribosome footprint profiling library preparation**. HEK293 cells transiently transfected with 4× ACE-tRNAs and control plasmid (puc57GG) were grown in standard grown media in the absence of Pen-Strep for 48 h. Libraries were prepared as described[61], with a few modifications. Briefly, cells were rapidly cooled by addition of ice-cold PBS, lysed in lysis buffer (20 mM Tris-HCl/pH7.4, 150 mM NaCl, 5 mM MgCl$_2$, 1 mM DTT, 1% (v/v) Triton X-100, and 25 U ml$^{-1}$ Turbo DNase I) for 10 min on ice, and triturated with ten times through a 26-G needle. After clearance by centrifugation at 16,000×g for 10 min at 4 °C, the lysates were digested with 100 U RNase I (Ambion, USA) per A$_{260}$ lysate at room temperature for 45 min with gentle agitation prior to adding 200 U RiboLock RNase Inhibitor (Thermo Scientific). Ribosome protected mRNA fragments were then isolated by loading lysates onto a 1 M sucrose cushion prepared in modified polysome buffer (20 mM Tris-HCl/pH 7.4, 150 mM NaCl, 8.5 mM MgCl$_2$, 0.5 mM DTT, 20 U ml$^{-1}$ RiboLock RNase Inhibitor) and centrifuged at 70,000 rpm at 4 °C for 2 h using a Beckman TLA-110 rotor. Ribosome pellets containing mRNA footprints were extracted using TRIzol and separated on a denaturing 12% polyacrylamide gel containing 8 M urea. RNA fragments with sizes ranging from 26 to 34 nt were manually excised from the gel stained with SYBR Gold (Invitrogen) and isolated to generate the ribosome-protected fragment library. Contaminating rRNA fragments depleted using a Ribo-Zero kit (Illumina). 3′ Oligonucleotide adapter ligation, reverse transcription, circularization, and secondary rRNA depletion using biotinylated rRNA depletion oligos (Supplementary Information, Supplementary Table 1) were performed as described[61]. Libraries were barcoded using indexing primers for each sample during PCR amplification. Barcoded libraries were then pooled with 3% PhiX (Illumina) and sequenced in an Illumina NextSeq 500 as per manufacturer protocol to typically generate 18–27 million reads per sample.

**Ribosome footprint data analysis**. Data files for each barcoded sample (minus adapter sequence at 3′ end) were first mapped to four rRNA sequences (RNA5S1; NR_023363, RNA5-8SN5; NR_003285, RNA18SN5;NR_003286, and RNA28SN5; NR_003287) using HISAT 2.0.3[62] to eliminate rRNA contaminant reads. The remaining reads were aligned to the sense stands of the longest transcript variant of each human gene (UCSC RefSeq GRCh38). Transcripts with 3′UTR length of at least 75 nt (18,101 sequences) were used for subsequence analysis. A maximum of two mismatches at the 5′end of reads was allowed. All multi-mapped reads were discarded. Fragment reads with lengths between 26 and 34 nt were defined as ribosome footprints and used for analysis. The 5′ end nucleotide from each footprint was annotated and mapped on each transcript. Position of the ribosome A-site occupying the 16th–18th nucleotides of each footprint[63,64] was used to infer the position of the ribosome on each transcript. RPKM (footprint Reads Per Kilobase of transcript per total Million-mapped reads) on each individual transcript (18,101 sequences) was calculated. Only transcripts with a minimum threshold of 5 RPKM in the coding sequence and 0.5 RPKM in 3′UTR region in two replicate libraries (254 transcripts in G418 and 495-748 transcripts in ACE-tRNAs) were included for analysis in Fig. 4a. For transcriptome-wide metagene

plots in Fig. 4b, footprint counts for each nucleotide within the region from +6 to +65 nt relative to the first nucleotide of stop codon were normalized per total million-mapped reads. All transcripts (18,101 sequences) were used for mapping, and more than 2313 transcripts were mapped to at least 1 footprint in the region of interest. The sequencing data was analyzed using Galaxy platform[65]. Graphs were generated using Prism 7 (GraphPad Software).

**Generation of stable NLuc reporter cell lines**. The cDNAs encoding pNLuc with tag, taa and tga stop codons at amino acid position 160 were inserted into AgeI and NotI restriction sites within the multiple cloning site of the retroviral vector pQCXIP (Clontech, USA) using Gibson Assembly (New England Biolabs, USA). PhoenixGP cells[66] were co-transfected with pNLuc-STOP-pQCXIP and cmv-VSV-G (VSV-G envelope pseudotyping) plasmids using Calfectin (SignaGen Laboratories, USA) and placed in a 33 °C CO$_2$-controlled (5%) cell incubator for 48 h. The culture media (20mls) containing retroviral particles was chilled to 4 °C and spun at 10,000×g to remove cell debris and filtered through a 0.45 μm MCE-membrane syringe filter (Millipore, USA) onto two 10 cm dishes seeded with low-passage HEK293 cells at 30% confluency. Cell culture dishes were sealed with Parafilm and spun for 90 min at 3500×g at 24 °C and placed in a 37 °C CO$_2$ controlled (5%) cell culture incubator. Cells were selected 24 h later with puromycin (1 μg/ml) until the control dish (no infection) showed complete cell death. Cells were monodispersed into 96-well plates using FACS and clonal populations were subsequently. Puromycin was not used to maintain selected clones during experimentation and standard DMEM media (Dulbecco's Modified Eagle Medium-high glucose with L-glutamine supplemented with 10% FBS, 1% Pen/Step and 2mM L-Glutamine; ThermoFisher, USA) was used in all studies.

**RNA transfection**. HEK293 cells stably expressing pNLuc-UGA were plated at $1.4 \times 10^4$ cells/well in 96-well cell culture treated plates in DMEM supplemented with 10% FBS, 1% Pen/Step and 2 mM L-Glutamine (Thermofisher, USA). 16–24 h later the cells were transfected with ACE-tRNAs using lipofectamine 2000 (ThermoFisher Scientific, USA). Briefly, 3 μg of ACE-tRNA were suspended in 150 μl of OptiMEM and 12 μl of Lipofectamine 2000 was mixed with 150 μl of OptiMEM. The volumes were combined, thoroughly mixed and incubated for 10 min at RT. Seventy-five microliter of the transfection complex was added to each well. PTC suppression by ACE-tRNA transcripts was quantified as described above.

**Expression in *Xenopus laevis* oocytes**. *Xenopus laevis* oocytes (stage V and VI) were purchased from Ecocyte (Austin, TX). Prior to injection, each ACE-tRNA pellet was resuspended in 2 μl of ddH$_2$O and debris was pelleted at 21,000×g, 4 °C for 25 min. To determine dose response of ACE-tRNAs on CFTR channel rescue, we generated serial dilutions of ACE-tRNA aliquots (200, 100, 50, 25, 12.5, 6.25, 3.125, and 1.562 ng/oocyte) balanced in volume with ddH$_2$O. In all experiments 25 ng of *CFTR* cRNA was injected per oocyte and injection volumes were 50 nl. ddH$_2$O was used in no ACE-tRNA background control experiments. After injection, oocytes were kept in OR-3 (50% Leibovitz's medium, 250 mg/l gentamycin, 1 mM L-Glutamine, 10 mM HEPES (pH 7.6)) at 18 °C for 36 h.

**Two-electrode voltage clamp (TEVC) recordings**. CFTR Cl$^-$currents were recorded in ND96 bath solution that contained (in mM): 96 NaCl, 2 KCl, 1 MgCl$_2$, and 5 HEPES (pH 7.5) in the presence of a maximal CFTR activation cocktail, forskolin (10 μM; adenylate cyclase activator) and 3-isobutyl-1-methylxanthine (1 mM; phosphodiesterase inhibitor). Glass microelectrodes backfilled with 3 M KCl had resistances of 0.5–2 MΩ. Data were filtered at 1 kHz and digitized at 10 kHz using a Digidata 1322 A controlled by the pClamp 9.2 software (Molecular Devices, USA). *CFTR* currents were elicited using 5 mV voltage steps from −60 to +35 mV using an OC-725C voltage clamp amplifier (Warner Instruments, USA). Oocytes where the CFTR Cl$^-$ current reversed positive of −20 mV were discarded. Clampfit 9.2 software was used for current analysis. All values are presented as mean ± SEM.

**Animals and in vivo imaging**. Nu/J mice were purchased from Jackson labs. Animal experiments were approved by the Institutional Animal Care and Use Committee at the Wistar Institute (protocol number: 112762). Mice were treated by injecting 10–20 μg of water into the tibialis anterior muscle followed by electroporation. We injected 10 μg pNLuc-UGA+10 μg 4xACE-tRNA$^{Arg}$ (right tibialis anterior) or 10 μg pNLuc-UGA+10 μg empty pUC57 (left tibialis anterior) to 3 mice. As controls we injected three other mice with 10 μg pNLuc-WT (right tibialis anterior; positive control) or water (left tibialis anterior; negative control). The DNA was formulated with 333IU/ml of hyaluronidase (Sigma). One minute after DNA injection we proceeded to electroporation with CELLECTRA 3P device (Inovio Pharmaceuticals). We imaged nanoluciferase activity in mice by injecting 100 μl of furimazine (40× dilution of Nano-Glo substrate) intraperitoneally and imaged mice on an IVIS Spectrum (Perkin Elmer) 5 min after injection. We imaged with open filter and acquired images at 40 s. We analyzed the images using Living Image Software (Perkin Elmer).

**Reporting Summary**. Further information on experimental design is available in the Nature Research Reporting Summary linked to this Article.

## Data availability

Sequencing data has been uploaded to Mendeley under the https://doi.org/10.17632/mgtfjbxxtf.1 (https://doi.org/10.17632/mgtfjbxxtf.1). The mass spectrometry proteomics data have been deposited to the ProteomeXchange Consortium via the PRIDE[67] partner repository with the dataset identifier PXD012581. All other data are available within the manuscript and accompanying materials or from the authors upon reasonable request.

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

## Acknowledgements

This work and JDL were supported by Emily's Entourage (www.emilysentourage.org). Additional support was provided by a Cystic Fibrosis Foundation (CFF) Pilot Award (R458-CR11) and Research Grant (498721), the NIH (GM106568) and CAA is an American Heart Association Established investigator (5EIA22180002). J.S.-Y. and W.R.S. are supported directly through the CFF and Cystic Fibrosis Foundation Therapeutics (CFFT) Research laboratory. DTI is supported by the Cystic Fibrosis Foundation (INFIEL17F0). D.B.W. is funded by the WW Smith Family Trust. PBM and CAA are supported by the Roy J. Carver Charitable Trust. The authors acknowledge formative discussions with J. Kevin Foskett (University of Pennsylvania) that facilitated development of this study. We thank Dr. Shikha Mishra at ThermoFisher for fruitful nucleotide delivery discussion and providing reagents used in this study. We thank Dr. Julien Sebag (University of Iowa, Department of Molecular Physiology and Biophysics) for providing access to Spectramax i3. We thank Grace Galles and Selena Borrill for their technical assistance. We acknowledge Inovio Pharmaceuticals for the use of the CELLECTRA 3P for in vivo electroporation. Approved by Institutional Animal Care and Use Committee at the Wistar Institute (protocol number: 112762).

## Author contributions

J.D.L., M.A.B., D.B.W., W.R.S., P.B.M, and C.A.A. designed the study. J.D.L., J.S.Y., A.P-P., A.L.M., D.T.I., and M.R.P. performed experiments, analyzed the data, and constructed the figures. J.D.L. and C.A.A. wrote the manuscript. All authors read and revised the manuscript.

## Additional information

**Competing interests:** D.B.W. receives research funding from Inovio Pharmaceuticals, and from GeneOne Pharmaceuticals. He has received Honaria for speaking at Merck, Roche & AstraZeneca, has ownership interest (including patents) in Inovio Pharmaceuticals and has been a consultant/advisory board member for Inovio Pharmacueticals and Gene One ((PCT/US2018/059065, filed November 2, 2018; METHODS OF RESCUING STOP CODONS VIA GENETIC REASSIGNMENT WITH ACE-tRNA; Inventors - University of Iowa - C.AA. and J.D.L.; Pertains to the tRNA sequences in Figure 2, Supplementary Figure 2a and Supplementary Data 1 and Data 2); (PCT/US2018/59085, filed November 2, 2018; METHODS OF RESCUING STOP CODONS VIA GENETIC REASSIGNMENT WITH ACE-tRNA; Inventors - The Wistar Institute of Anatomy and Biology, University of Iowa - A.P.-P., J.D.L., D.B.W. and C.A.A.; Pertains to in vivo delivery data shown in Figure 5. The remaining authors declare no competing interests.

