## [Peer Review File · Nature Communications]

Reviewers' comments:

Reviewer #1 (Remarks to the Author):

This manuscript "Engineered transfer RNAs for suppression of premature termination codons" by Lueck et al., focuses on identifying and understanding how ACE-tRNAs can suppress in frame PTCs and encode their cognate amino acid. This is an important question as PTCs are responsible for 10-15% of all inherited disease and to date there is no proven therapy for patients with PTCs. Since there are multiple tRNA genes with unique sequences for a given cognate amino acid (over 400 tRNA genes to date), the group generated a cDNA plasmid that incorporated a high-throughput cloning strategy for the ACE-tRNAs and quantitative suppression of PTC using luminescence once delivered into cells. They focused on screening codon-edited tRNA for each of the possible single nucleotide mutation which would produce disease causing PTCs. They found multiple ACE-tRNAs in each screen that successfully suppressed the PTCs in robust fashion. They showed through mass spectrometry that the ACE-tRNAs were working as expected and were quite efficient at substituting the cognate amino acid. Importantly, they showed that the ACE-tRNAs did not substantially suppress native stop codons which is always a major question when PTC suppression is utilized. They observed in vivo activity and efficacy of these ACE-tRNAs to suppress PTCs in the muscle of mice in which plasmids encoding a reporter and ACE-tRNAs were delivered. Finally, they used the *Xenopus leavis* oocyte system to show that ACE-tRNAs can suppress specific disease causing CFTR PTCs (G542X and W1282X) to produce CFTR function.

The major findings of the paper are: 1) The group successfully identified multiple ACE-tRNAs that suppress three different PTCs, 2) These ACE-tRNAs efficiently inserted cognate amino acids and in the current systems produced better suppression than G418, 3) ACE-tRNAs do not substantially suppress native stop codons, 4) ACE-tRNAs can suppress disease causing mutations and produce functional protein and may be able to be used in vivo depending on the delivery and tissue type. These findings are novel and will be of great interest to the scientific community.

Comments:

1) It would have been nice to see more in vivo work with disease specific animal models in this manuscript however the amount of work already contained in this manuscript is immense and can stand on its own. An addition to the discussion of the next step in disease animal models may be useful.

2) Experiments and results described in Lines 136-143. The rationale for these experiments needs work. The authors state that they completed the experiments using tRNA sequences from other species because they wanted to expand UGA suppressing Trp ACE-tRNAs. However, they didn't find any successful suppression from ACE-tRNAs from other species. May need to explain why this is important. Could they test other ACE-tRNAs from these species that did suppress UGA like ACE-tRNALys? Not sure the conclusion you can make from this with no success and none others tried. Seems a little out of place.

3) Experiments and results in lines 174-196. The experiments are sound and the explanation is solid. I agree with the authors that most likely the readthrough that is observed from Gln-UAA and Arg-UGA ACE-tRNAs is very low and may not be physiological relevant. Is there any positive control to put in these systems to show what substantial native codon readthrough would look like?

4) Statement in lines 206-207. "Furthermore, this duration and intensity of luciferase expression argues in favor of a high in vivo tolerability and negligible repercussion of increased readthrough observed with ACE-tRNAArg." Be careful in this wording. Just because it is in muscle doesn't mean it will be tolerated in whole body or other organs. As you state later, this may be dependent on delivery and tissue type.

Minor comments:

1) The sentence (line 72-74)- "This is not limited to the requirement of "precision" or "personalized" diagnostics for each mutation based on the context of each patient's genetic variability." Please define "this". The sentence is unclear if the writer is referring to the technology or something else.

2) Figure 1 figure legend describes cognate amino acid as green circle but it is blue. Please change.

3) There is a different naming system between supplemental figure 2 and 3. It would be useful to know which of the 21 ACE-tRNAs in sup. Figure 3 corresponds to the 3 successful suppressors (17-19 in sup. Figure 2).

4) "Many of these ACE-tRNAs exhibited strong activity with >100-fold PTC suppression over background, which is significantly higher than the aminoglycosides used in this study." Line 120-122. Since the aminoglycoside data are shown after this you might want to add (see below) as readers may think they missed this data.

5) Line 153- Supplemental Figure 4b does not exist. Please change.

Reviewer #2 (Remarks to the Author):

This manuscript describes the creating of a large library of ACE-tRNA species that can read stop codons and suppress premature termination mutations associated with human genetic diseases. The authors identified several ACE-tRNAs for each amino acid and showed in specific cases that some of these ACE-tRNAs suppress premature termination codons in vitro, in cells and in animal models. The identified ACE-tRNAs are of interest to the field of gene therapy and highlight the potential that tRNA species can serve as therapeutic agents. The overall data are promising and provide encouragement to move forward with this approach. However, the following points need to be addressed.

In the Introduction section, "However, aspects of this technology (CRISPR/Cas9) impart hurdles for its rapid use as a therapeutic". The hurdles of the CRISPR/Cas should be briefly stated.

In the Results section, "This strategy (cloning of DNA oligs to the HTC plasmid using the golden gate cloning paired with ccdB negative selection) produced ~100% cloning efficiency. The rationale for producing ~100% efficiency should be described. What is the negative selection and how does it make ~100% success?"

In the Results section, "Overall, the tRNA screens identified multiple engineered tRNAs for each amino acid and stop codon type)...". It is clear that the screens produced multiple tRNAs for each amino acid. It is not clear if each amino acid has tRNAs with different anticodon types. It appears that each amino acid has only one anticodon type. Please clarify. If this is true, then the rationale for which tRNA anticodon type is chosen for each amino acid should be given.

In the Results section, "A TGA codon was introduced at N94 and co-expressed in HEK293 cells in tandem with plasmids encoding Glychr19.trna2.....". "In tandem" means "one after another". But in this case, it appears that there were two plasmids in each cell, one harboring the reporter and the other the tRNA gene. If so, then the wording should be changed.

A key question concerning the data is the stability of the ACE-tRNAs. When the production high reporter expression was reported, it is not known whether it was due to the stability of the tRNA or the reporter protein? This question refers not only to reporter assays, but also to experiments using directly tRNAs as the transfection agents.

Figure 5c, the effect of ACE-tRNA rescue by either plasmid expression or direct transfection with tRNA should be directly compared in one graph. Also, Figure 5c has no error bars.

For broader impact of the work for the tRNA community, each ACE-tRNA that is of the highest functional activity and suppression efficiency should be shown in a cloverleaf structure. Only the top performer for each amino acid is needed.

In the Discussion section, "Notably, our screen identifies ACE-tRNA, in total, with the ability to repair a vast majority of known human disease-causing PTC". This is a grossly over-statement. The majority of ACE-tRNAs have not been shown with the ability to repair a vast majority of known human disease-causing PTC.

The suppression of the UGA stop codon should include a discussion on selenocysteine tRNA, which reads the UGA stop codon but requires a specialized mRNA secondary structure. The possibility of whether the created ACE-tRNA might interfere with Selenocysteine insertion should be discussed.

As far as I can tell, no Tables are present in the manuscript, but Tables are mentioned in the text.

Figure 4, only 5 ACE-tRNAs were analyzed for read-through of the natural stop codons. Why these 5 tRNAs were chosen for the analysis? What about the ACE-tRNAs for the remaining 15 amino acids?

Supplementary Figure 1. Red highlight indicates "the most common codons" and corresponding amino acid....How are the most common codons defined?

This manuscript “Engineered transfer RNAs for suppression of premature termination codons” by Lueck et al., focuses on identifying and understanding how ACE-tRNAs can suppress in frame PTCs and encode their cognate amino acid. This is an important question as PTCs are responsible for 10-15% of all inherited disease and to date there is no proven therapy for patients with PTCs. Since there are multiple tRNA genes with unique sequences for a given cognate amino acid (over 400 tRNA genes to date), the group generated a cDNA plasmid that incorporated a high-throughput cloning strategy for the ACE-tRNAs and quantitative suppression of PTC using luminescence once delivered into cells. They focused on screening codon-edited tRNA for each of the possible single nucleotide mutation which would produce disease causing PTCs. They found multiple ACE-tRNAs in each screen that successfully suppressed the PTCs in robust fashion. They showed through mass spectrometry that the ACE-tRNAs were working as expected and were quite efficient at substituting the cognate amino acid. Importantly, they showed that the ACE-tRNAs did not substantially suppress native stop codons which is always a major question when PTC suppression is utilized. They observed in vivo activity and efficacy of these ACE-tRNAs to suppress PTCs in the muscle of mice in which plasmids encoding a reporter and ACE-tRNAs were delivered. Finally, they used the *Xenopus leavis* oocyte system to show that ACE-tRNAs can suppress specific disease causing CFTR PTCs (G542X and W1282X) to produce CFTR function.

The major findings of the paper are: 1) The group successfully identified multiple ACE-tRNAs that suppress three different PTCs, 2) These ACE-tRNAs efficiently inserted cognate amino acids and in the current systems produced better suppression than G418, 3) ACE-tRNAs do not substantially suppress native stop codons, 4) ACE-tRNAs can suppress disease causing mutations and produce functional protein and may be able to be used in vivo depending on the delivery and tissue type. These findings are novel and will be of great interest to the scientific community.

We thank the reviewer for their thorough reading of the manuscript and for highlighting the relevance of the data. We have responded to each comment below and subsequent changes to the manuscript are noted in a case by case manner.

Comments:

1) It would have been nice to see more in vivo work with disease specific animal models in this manuscript however the amount of work already contained in this manuscript is immense and can stand on its own. An addition to the discussion of the next step in disease animal models may be useful.

We agree with the reviewer's sentiment that the next logical step for the work is to address the application of the ACE tRNA data to PTC disease specific cellular and amino models. At the same time, we appreciate that the reviewer considered the amount of data within the paper is already highly significant "as is" and will be of value for the scientific community. Overall, we feel that our dataset successfully advances the concept of a suppressor tRNA to an approach with bona fide therapeutic potential. To help frame the next steps for the reader, we added a brief paragraph to the Discussion outlining the likely challenges in advancing the technology to disease specific models.

Lines 281-287: "One of the factors that has hampered PTC suppression therapeutics is the dearth of sufficient nonsense-associated animal models of disease and/or PTC suppression reporter animals. Using the rapid and precise CRISPR/Cas mouse genome manipulation methodologies, generation of appropriate nonsense-associated animal models of disease will greatly facilitate advances in PTC therapeutics centered around traditional small molecules (i.e aminoglycoside chemistries), nuclease technologies (i.e. CRISPR/Cas and TALENS) and ACE-tRNA based approaches described here."

Given the limited therapeutic options for the PTC community, we hope that this discussion spurs the interests of other scientists with expertise in specific PTC diseases and RNA delivery.

2) Experiments and results described in Lines 136-143. The rationale for these experiments needs work. The authors state that they completed the experiments using tRNA sequences from other species because they wanted to expand UGA suppressing Trp ACE-tRNAs. However, they didn't find any successful suppression from ACE-tRNAs from other species. May need to explain why this is important. Could they test other ACE-tRNAs from these species that did suppress UGA like ACE-tRNA^{Lys}? Not sure the conclusion you can make from this with no success and none others tried. Seems a little out of place.

We thank the reviewer for this helpful comment and agree with this sentiment. As the reviewer has surmised, the motivation for these additional Trp screens was to improve the chances to identify ACE Trp sequences with improved UGA tolerance and suppression ability. As these efforts were largely unsuccessful, we agree that they are somewhat out of place in the current study. None the less, we prefer to keep them in the Supplemental (SFig. 5) as they will be of value to the community for the understanding of suppressor Trp tRNA.

3) Experiments and results in lines 174-196. The experiments are sound and the explanation is solid. I agree with the authors that most likely the readthrough that is observed from Gln-UAA and Arg-UGA ACE-tRNAs is very low and may not be physiological relevant. Is there any positive control to put in these systems to show what substantial native codon readthrough would look like?

The mechanism of PTC readthrough remains poorly understood thus the choice for an "ideal" positive control is not straight forward. However, we compared the standard readthrough agent G418 with ACE tRNA interactions with protein termination codons. This aminoglycoside has an established record in the literature for promoting PTC readthrough. In our hands, the biological activity of this particular agent was modest, but serves as an internal readthrough control that would be accepted by most members of the PTC community. In the future, a potentially better positive control could be a cellular

assay in which some aspects of the termination complex (RF1, etc) were genetically disabled. Such efforts are underway for future studies and for investigations of termination suppression mechanisms.

4) Statement in lines 206-207. “Furthermore, this duration and intensity of luciferase expression argues in favor of a high in vivo tolerability and negligible repercussion of increased readthrough observed with ACE-tRNA^{Arg}.” Be careful in this wording. Just because it is in muscle doesn’t mean it will be tolerated in whole body or other organs. As you state later, this may be dependent on delivery and tissue type.

Thank you for this cautionary suggestion as we do not wish to oversell the data. We have modified the sentence to now read as follows:

Lines 212-215: “Furthermore, this duration and intensity of luciferase expression is generally supportive of in vivo tolerability with ACE-tRNA^{Arg}. However, additional experimentation will be needed to acquire delivery dependent expression and to identify any potential tissue specific tolerance issues.”

Minor comments:

1) The sentence (line 72-74)- “This is not limited to the requirement of “precision” or “personalized” diagnostics for each mutation based on the context of each patient’s genetic variability.” Please define “this”. The sentence is unclear if the writer is referring to the technology or something else.

We have clarified this section by combining with the previous sentence. This segue now reads:

Lines 70-74: “However, aspects of this technology impart hurdles for its rapid use as a therapeutic^{23,24}, and the challenges for advancing gene editing are not limited to the requirement of “precision” or “personalized” diagnostics for each mutation based on the context of each patient’s genetic variability.”

2) Figure 1 figure legend describes cognate amino acid as green circle but it is blue. Please change.

Corrected.

3) There is a different naming system between supplemental figure 2 and 3. It would be useful to know which of the 21 ACE-tRNAs in sup. Figure 3 corresponds to the 3 successful suppressors (17-19 in sup. Figure 2).

We thank the reviewer for pointing out this issue. We have now numbered the tRNAs in all supplemental figures. Importantly, the ACE-tRNAs within the amino acid families can now be indexed against their “activity” in Table 2. of the supplemental figures.

4) “Many of these ACE-tRNAs exhibited strong activity with >100-fold PTC suppression over background, which is significantly higher than the aminoglycosides used in this study.” Line 120-122. Since the aminoglycoside data are shown after this you might want to add (see below) as readers may think they missed this data.

Thank you for this suggestion. We have added “(see below)” to this sentence.

5) Line 153- Supplemental Figure 4b does not exist. Please change.

This refers to Supplemental Figure 6 and has been changed accordingly.

Reviewer #2 (Remarks to the Author):

This manuscript describes the creating of a large library of ACE-tRNA species that can read stop codons and suppress premature termination mutations associated with human genetic diseases. The authors identified several ACE-tRNAs for each amino acid and showed in specific cases that some of these ACE-tRNAs suppress premature termination codons in vitro, in cells and in animal models. The identified ACE-tRNAs are of interest to the field of gene therapy and highlight the potential that tRNA species can serve as therapeutic agents. The overall data are promising and provide encouragement to move forward with this approach. However, the following points need to be addressed.

We thank the reviewer for their careful read of our paper and we are heartened that they find data are supportive of the therapeutic potential for the approach.

In the Introduction section, "However, aspects of this technology (CRISPR/Cas9) impart hurdles for its rapid use as a therapeutic". The hurdles of the CRISPR/Cas should be briefly stated.

This section has been updated with a brief description of the CRISPR/Cas therapeutic hurdles, see above description and updated content in Line 67-75.

In the Results section, "This strategy (cloning of DNA oligs to the HTC plasmid using the golden gate cloning paired with ccdB negative selection) produced ~100% cloning efficiency. The rationale for producing ~100% efficiency should be described. What is the negative selection and how does it make ~100% success?"

We agree that this section should be elaborated upon and have now included a more sufficient rationale for our molecular biology methods in Lines 102-105.

In the Results section, "Overall, the tRNA screens identified multiple engineered tRNAs for each amino acid and stop codon type)...". It is clear that the screens produced multiple tRNAs for each amino acid. It is not clear if each amino acid has tRNAs with different anticodon types. It appears that each amino acid has only one anticodon type. Please clarify. If this is true, then the rationale for which tRNA anticodon type is chosen for each amino acid should be given.

The rationale for the tRNA chosen for codon editing tolerance is based primarily on their role in human disease. Specifically, tRNA with cognate suppression codons one nucleotide substitution from a stop codon (TAA, TAG, TGA) were examined. Given the degeneracy of the genetic code, multiple codons exist for each amino acid, and in terms of PTC prevalence, the consequence of this degeneracy is that, in some cases, a single amino acid triplet codon can become more than one stop codon type. This was the case for Glutamate which from GAA and GAG could become TAG or TAA. Similar examples include: Glutamine (TAA/TAG), Tryptophan (TGA/TAG), and Lysine (TGA/TAG). In the case of Cysteine, two of its cognate codons (TGC, TGT) could be converted to a single stop codon type, TGA. Whereas Leucine, with six cognate codons, can be converted to TAG, TGA, and TAA.

In the Results section, "A TGA codon was introduced at N94 and co-expressed in HEK293 cells in tandem with plasmids encoding Glychr19.trna2.....". "In tandem" means "one after another". But in this case, it appears that there were two plasmids in each cell, one harboring the reporter and the other the tRNA gene. If so, then the wording should be changed.

Thank you for pointing out this potential source of confusion. "In tandem" has been replaced with "together".

A key question concerning the data is the stability of the ACE-tRNAs. When the production high reporter expression was reported, it is not known whether it was due to the stability of the tRNA or the reporter protein? This question refers not only to reporter assays, but also to experiments using directly tRNAs as the transfection agents.

Thank you for making this important point. To address this specific comment, we now include new data showing the expression profiles obtained with transfected tRNA as RNA (Supplementary Fig. 11). These data show that the RNAs can support suppression up to 2 days, after which reporter signal returns to baseline.

Figure 5c, the effect of ACE-tRNA rescue by either plasmid expression or direct transfection with tRNA should be directly compared in one graph. Also, Figure 5c has no error bars.

We thank the reviewer for this suggestion. We performed a head-to-head comparison of cDNA and RNA suppression activity in HEK293 cells, and these results are in Supplemental Fig. 11. We added the following text to the results section.

Lines 219-222: "In separate experiments, we compared the duration of ACE-tRNA rescue delivered as RNA and cDNA to HEK293 cells (Supplemental Fig. 11). ACE-tRNA activity delivered as RNA peaked at 8hrs following delivery and decreased over a 48hr period, whereas cDNA expression plasmids supported an increase in PTC suppression activity that plateaus at 48hrs."

Figure 5c (Trp) suppression data are from three experiments with similar value, thus the error bars plotted as S.E. are highly compressed.

For broader impact of the work for the tRNA community, each ACE-tRNA that is of the highest functional activity and suppression efficiency should be shown in a cloverleaf structure. Only the top performer for each amino acid is needed.

We agree and now have added clover leaf structures of all of the top performer tRNA sequences to the supplemental figures as Supplemental Fig. 2B that are projected by tRNAscan-SE. The non-engineered sequences were used for the structure generation, and mutation of the anticodon does not affect the structures.

In the Discussion section, "Notably, our screen identifies ACE-tRNA, in total, with the ability to repair a vast majority of known human disease-causing PTC". This is a grossly over-statement. The majority of ACE-tRNAs have not been shown with the ability to repair a vast majority of known human disease-causing PTC.

It was not our intention to overstate the significance of the data but only to highlight the potential therapeutic utility of the identified tRNA sequences and expression data. We have adjusted the sentence accordingly:

Lines 266-268: "Notably, our screen identifies ACE-tRNA, in total, with the potential to repair a vast majority of known human disease-causing PTC, but this therapeutic will require overcoming tissue and delivery specific challenges."

The suppression of the UGA stop codon should include a discussion on selenocysteine tRNA, which reads the UGA stop codon but requires a specialized mRNA secondary structure. The possibility of

whether the created ACE-tRNA might interfere with Selenocysteine insertion should be discussed.

The thank the reviewer for pointing out this important point. We have added a brief discussion of the selenocysteine encoding complex in the Discussion, lines 302-304.

As far as I can tell, no Tables are present in the manuscript, but Tables are mentioned in the text.

Two tables are included in the resubmitted manuscript. Table 1 lists the tRNA sequences and Table 2 contains the statistical comparisons of the luciferase rescue data.

Figure 4, only 5 ACE-tRNAs were analyzed for read-through of the natural stop codons. Why these 5 tRNAs were chosen for the analysis? What about the ACE-tRNAs for the remaining 15 amino acids?

The ribosomal profiling assays are resource intensive, which limited our ability to include more tRNA subtypes. For this reason, we focused the analysis on tRNA suppressors with a broad range of PTC rescue (fold over background) and stop codon suppressor type. This approach allowed us to compare readthrough at two stops codons (TAA and TAG) from as single tRNA type, Gln. In comparison, TGA readthrough was analyzed for ACE-Arg, ACE-Gly, and ACE-Trp; tRNAs with a broad range (10-fold to 1000-fold) of rescue abilities. Efforts to quantify the effects of other identified ACE-tRNA are underway and part of future work.

Supplementary Figure 1. Red highlight indicates "the most common codons" and corresponding amino acid....How are the most common codons defined?

Thank you for pointing this out to us. The codons highlighted in red are the codons to result in a PTC that are only one nucleotide substitution away from a termination codon, and therefore the most common. The mutagenesis rate of the human genome is so low (3.0×10^{-8} mutations/nucleotide/generation; Xue et al., 2009), therefore the likelihood of getting multiple nucleotides substituted to generate a PTC would be extremely rare.

In closing, we believe the impact of this work is highly significant to several fields, including: tRNA biology, protein translation and termination, genetic-code expansion, protein function and physiology. Moreover, from the perspective of health and disease the data describe an entirely new class of RNA biologicals that hold tremendous potential for treating of human disease.